# Visual Semantic Navigation using Scene Priors

**Wei Yang**[1]**, Xiaolong Wang**[2]**, Ali Farhadi**[4,5]**, Abhinav Gupta**[2,3]**, Roozbeh Mottaghi**[5]
[1] The Chinese University of Hong Kong [2] Carnegie Mellon University [3] Facebook AI Research
[4] University of Washington [5] Allen Institute for AI

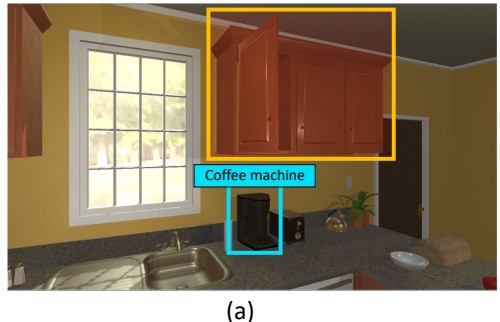 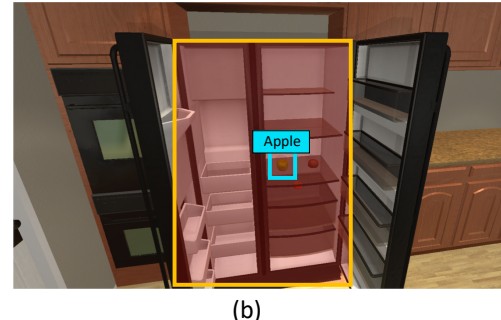

(a)  (b)

Figure 1: Our goal is to use scene priors to improve navigation in unseen scenes and towards novel objects. (a) There is no mug in the field of view of the agent, but the likely location for finding a mug is the cabinet near the coffee machine. (b) The agent has not seen a mango before, but it infers that the most likely location for finding a mango is the fridge since similar objects such as apple appear there as well. The most likely locations are shown with the orange box.

## Abstract

How do humans navigate to target objects in novel scenes? Do we use the semantic/functional priors we have built over years to efficiently search and navigate? For example, to search for mugs, we search cabinets near the coffee machine and for fruits we try the fridge. In this work, we focus on incorporating semantic priors in the task of semantic navigation. We propose to use Graph Convolutional Networks for incorporating the prior knowledge into a deep reinforcement learning framework. The agent uses the features from the knowledge graph to predict the actions. For evaluation, we use the AI2-THOR framework. Our experiments show how semantic knowledge improves performance significantly. More importantly, we show improvement in generalization to unseen scenes and/or objects.

## 1 Introduction

Consider the kitchen scene shown in Figure 1(a) and the task of finding an object such as a mug. Even though we have never seen this particular kitchen before and no mug is visible in the scene, we can still infer the likely locations to find the mug and create an exploration plan accordingly. For example, in Figure 1(a), we can infer that since there is a coffee machine, the mug is most likely in the cabinet near the coffee machine. How do we do that? We infer that mugs are usually used for coffee. And since there is a coffee machine, the mug is likely to be near the machine due to functional efficiency. We argue that humans use strong priors about the functional and semantic structure of the world to develop such efficient navigation strategies. And how do we learn such functional/semantic priors? Our prior experience and exploration of tens of kitchens help us to learn these priors.

But these priors are not just used for navigating to known objects but also to completely unknown and unseen objects. For example, let us assume you have never seen a mango before and someone

gives you a task of finding a mango in a new kitchen you have never seen before (let's say Figure 1(b)). How would you do it? Assuming you have searched for fruits like apples and grapes before, and you know mango is also a fruit; so a similar exploration strategy might apply. Therefore, in Figure 1(b), you are more likely to navigate to the fridge to search for a mango. Therefore, we use the semantic/functional priors to navigate to unseen objects as well.

Inspired by these observations, in this paper, we explore how to exploit semantic priors for the task of semantic and goal-oriented navigation. In our visual navigation task, the environment, the path to the target, the target location, or the exact appearance of the target object can be unknown. The prior knowledge about the semantic/functional structure of the world helps to improve the navigation. We propose to use Graph Convolutional Networks (GCNs) (Kipf & Welling, 2017) to incorporate the prior knowledge into a Deep Reinforcement Learning framework. The knowledge of the agent is encoded in a graph. GCNs allow arbitrary structured graphs to be encoded in an efficient way. The knowledge is updated according to the current observation of the agent, which is specific to the current environment, and the knowledge at the previous time step or the prior knowledge. The prior knowledge is obtained from large-scale datasets designed for scene understanding. Our model is based on the actor-critic model (Mnih et al., 2016) that is augmented by the knowledge graph and object visibility information.

To evaluate our model, we use the AI2-THOR framework (Kolve et al., 2017), which provides near photo-realistic customizable environments. The agent can take navigation actions in these environments and observe the changes as a result of those actions. AI2-THOR includes various objects that can be arranged in many different configurations. The agent location is randomized as well at each episode of training or testing. Our experiments show that the semantic prior improves the performance of the baseline RL models significantly. Furthermore, we show the results of the model on the challenging setting where the scene and/or the object are new to the agent.

Our contributions are summarized as follows: (1) We integrate a deep reinforcement learning model with knowledge graphs. This allows the agent to encode any form of knowledge that can be represented by graph structures. (2) We show that semantic prior knowledge can significantly improve the navigation performance. (3) By considering the prior information and the semantics of the target objects, we improve generalization to unseen environments and novel target objects.

## 2 RELATED WORK

Semantic and goal-oriented navigation is one of the most prominent tasks that intelligent species perform in their daily life. There are several challenges involved in visual navigation. First, the environment might be unknown to the agent. In this situation, the agent requires to explore the environment to have a better understanding of that environment. The second challenge is about the visibility of target objects. The target object might not be visible when the agent starts the navigation or it might go out of the field of view during navigation. Hence, the agent needs to learn an efficient search strategy to find the target object. The third challenge is related to planning. The object might be visible but planning a reasonable path towards the object is another issue that the agent needs to deal with. There have been several efforts in the past to tackle these challenges which we describe below.

**Geometry-based navigation.** Navigation methods can be divided into two main categories of geometry-based and learning-based. Most of the traditional navigation approaches fall into the former category, where it is assumed that either the map of the environment is known a priori, e.g., Matthies & Shafer (1987); Borenstein & Koren (1991); Meng & Kak (1993); Kim & Nevatia (1999) or a map is built on the fly e.g., Thrun (1998); Feder et al. (1999); Jones & Soatto (2011); Siagian et al. (2014). Our work is different from these approaches since we do not rely on a map for our navigation and we leverage semantic prior knowledge to reduce the required exploration time.

**Learning-based navigation.** Recent success of deep learning and reinforcement learning has made learning-based navigation approaches more popular. Zhu et al. (2017) propose a deep RL-based navigation approach, where they provide the picture of the target object. In contrast, we only provide semantic labels to the agent, so we can show generalization to unseen scenes. Gupta et al. (2017) propose a mapper and planner to output navigation actions. Mirowski et al. (2017) also propose a navigation framework that optimizes a loss for auxiliary tasks such as depth prediction and

loop closure classification. Sadeghi & Levine (2017) propose an RL-based approach for collision avoidance. Brahmbhatt & Hays (2017) explore a CNN-based approach for navigating in cities using local observations of the streets. Wu & Tian (2017) combine deep RL with curriculum learning in a first-person shooting game setting. Savinov et al. (2018) introduce a topological landmark-based memory for navigation. Kahn et al. (2018) propose a method based on model-free and model-based RL to learn navigation policies using a few samples. Mousavian et al. (2018) use object detection and semantic segmentation to better navigate in unseen environments. Chen et al. (2015) directly map the input image to an action in an autonomous driving setting. There is also a large body of work that address visually grounded navigation instructions e.g., Anderson et al. (2018b); Chaplot et al. (2018); Hermann et al. (2017); Yu et al. (2018); Misra et al. (2017); Mei et al. (2016). In contrast to all these approaches, we incorporate semantic and functional priors to improve navigation performance and better generalize to unseen scenes and objects.

**Context and scene prior.** Contextual reasoning has been studied extensively in the computer vision literature (Torralba et al., 2003; Hoiem et al., 2005; Rabinovich et al., 2005; Divvala et al., 2009; Desai et al., 2009; Marszalek et al., 2009; Malisiewicz & Efros, 2009; Mottaghi et al., 2014; Zhu et al., 2015; Shrivastava & Gupta, 2016). However, contextual information is mainly used for static settings such as object detection, semantic segmentation or action recognition. We use contextual reasoning for an interactive navigation task, where the agent updates its belief based on the current observation and the prior knowledge as it moves in the environment. Object relationships have been used for tasks such as image retrieval (Johnson et al., 2015), visual relation detection (Zhang et al., 2017), referring expressions (Nagaraja et al., 2016; Hu et al., 2017), and visual question answering (Johnson et al., 2017).

**Knowledge graphs.** There are recent works that use knowledge graphs for computer vision problems. A knowledge graph is used by Marino et al. (2017) for image classification, by Li et al. (2017) for situation recognition and by Wang et al. (2018) for zero-shot recognition. We use knowledge graphs in an RL setting for the interactive task of visual navigation.

**Reasoning about unknown environments or objects.** Various works have explored zero-shot reasoning in the context of reinforcement learning. Yu et al. (2018) address the problem of learning language in a 2D maze, where they can handle unseen word combinations or new sentences that contain unseen words. Harrison et al. (2017); Higgins et al. (2017) study zero-shot policy transfer in the scenarios that the dynamics or the states of the target domain is different from those of the source domain. Pathak et al. (2018) propose a zero-shot imitation learning approach where the expert demonstration for a particular task is never seen. Oh et al. (2017) address generalization of RL to unseen instructions and longer instructions. Our problem is different since we address navigation to novel objects or navigating in unseen scenes using *scene priors*.

## 3 VISUAL SEMANTIC NAVIGATION

In this section, we first define the task of visual semantic navigation. We then describe the formulation using deep reinforcement learning and the baseline model for the task.

### 3.1 TASK DEFINITION

Our goal is to navigate from a random starting location in a scene to a specified target object category given only egocentric RGB perception of the agent. The target object category is specified by a semantic label, thus we call our task visual semantic navigation. The task is considered successful if an instance of the target object category is visible. By "visible", we mean the target object is in the field of view and within a threshold of distance.

### 3.2 THE BASELINE MODEL

We formulate the visual semantic navigation using a deep reinforcement learning framework. Given a semantic task objective $g$, the agent perceives a state $s_t$ (i.e., the egocentric RGB image from the current location and orientation) at the time step $t$ and samples an action $a_t$ from the set of possible actions $\mathcal{A}$ according to its policy $\pi$. We approximate the policy by a deep policy network $\pi(\cdot; \theta)$:

$$a_t \sim \pi(\phi(s_t; u), \psi(g; v); \theta), \tag{1}$$

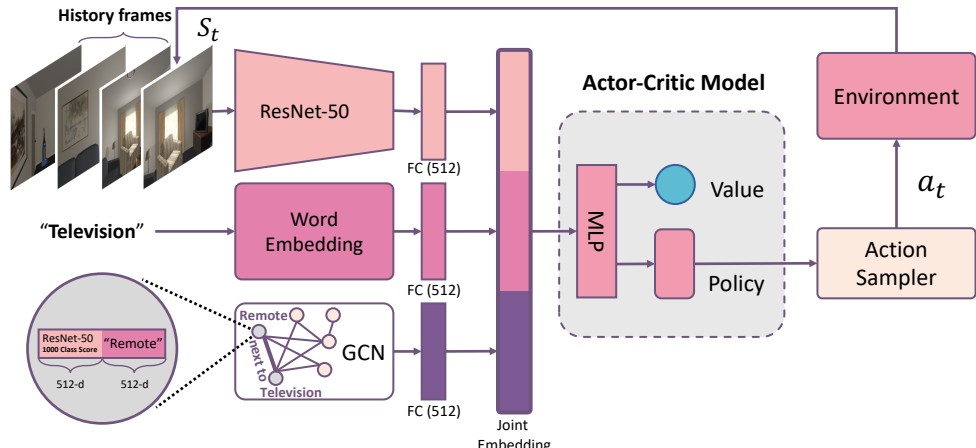

Figure 2: **Overview of the architecture.** Our model to incorporate semantic knowledge into semantic navigation. Specifically, we learn a policy network that decides an action based on the visual features of the current state, the semantic target category feature and the features extracted from the knowledge graph. We extract features from the parts of the knowledge graph that are activated.

where $u, v,$ and $\theta$ are the parameters for the network. Since the visual state and the semantic objective are from different modalities, we design two branches of subnetworks $\phi(\cdot; u)$ and $\psi(\cdot; v)$ to map these two inputs into a joint visual-semantic feature embedding.

**Visual network.** As illustrated in Figure 2 (top), the visual network takes $224 \times 224$ RGB images as input and generates a 512-d feature vector as output. The backbone of the visual branch is ResNet-50 (He et al., 2016) pre-trained on ImageNet. Specifically, we extract the 2048-d feature after the global average pooling of ResNet-50. To account for the history of the actions taken by the agent, we concatenate the features of the current frame and three past observations, which results in a 8192-d feature vector. We then add a fully connected layer and a ReLU layer to map the concatenated image feature into the 512-d visual-semantic feature.

**Semantic network.** The semantic task objective is described by an object category, e.g., *Microwave* or *Television*. We use fastText (Joulin et al., 2016) to compute a 100-d embedding for each word. Then we map the word embedding into a 512-d feature by a fully connected layer and ReLU, as illustrated in Figure 2 (middle).

**Actor-Critic policy network.** We employ the Asynchronous Advantage Actor-Critic (A3C) (Mnih et al., 2016) model to predict the policy at each time step. The input of our A3C model is the joint representation of the current state and the semantic task objective, which is a 1024-d feature vector made by concatenating the outputs of the visual network and the semantic network. The A3C model generates two outputs, i.e., the policy and the value. We sample the action from the predicted policy.

Our implementation of the A3C model consists of three layers: the input, the hidden layer, and the outputs. The hidden layer is a fully connected layer followed by the ReLU activation layer which maps the fused input into a 512-d latent space. Then the $|\mathcal{A}|$ dimensional policy and the value are generated by two branches of network, as shown in Figure 2. Unlike previous work (e.g., Zhu et al. (2017)) which uses different policy networks for different scenes, we use a single policy network for different scene examples. This makes our model more compact and generalizable.

**Reward.** We consider a reward to minimize the trajectory length to the targets: If any object instance from the target object category is reached within a certain number of steps, the agent receives a large positive reward 10.0. Otherwise, we penalize each step with a small negative reward -0.01. The design of the reward function is also affected by the types of actions $\mathcal{A}$. In our experiments, we ablate two sets of actions $\mathcal{A}$ with or without the *stop* action. In the setting without the *stop* action, the agent will receive the positive reward if the environment notifies it when it reaches the target, which also ends an episode of training. In the setting with the *stop* action, the episode is terminated when the *stop* action is executed, and the positive reward will be provided only if the agent is within the threshold of distance from the target (1 meter in our experiments) and facing the target. This makes the task much more challenging.

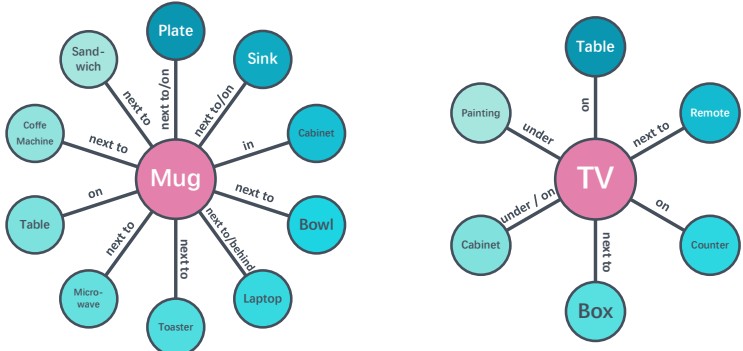

Figure 3: **Scene priors.** We extract relationships between objects from the Visual Genome (Krishna et al., 2017) dataset. The relationships for two example object categories are illustrated.

## 4 GENERALIZATION WITH GRAPH CONVOLUTIONAL NETWORKS

Our goal in this paper is to incorporate semantic knowledge into a Reinforcement Learning framework. To this end, we incorporate semantic knowledge in the form of graph representation and use Graph Convolutional Networks (GCNs) (Kipf & Welling, 2017) to compute relational features on the graph. GCNs allow us to incorporate prior knowledge and dynamically update it as the agent receives information specific to the current environment.

We first briefly describe how we build a semantic knowledge graph to represent the priors. We then provide the background for GCNs. Finally, we delve into the details of how we incorporate GCNs for the task of visual semantic navigation and how it helps generalization to unseen scenes and novel object categories.

### 4.1 KNOWLEDGE GRAPH CONSTRUCTION

Our knowledge graph for visual navigation provides two main advantages: (1) It encodes spatial relationships between different object categories. (2) It provides the spatial and visual relationships between the known objects and novel categories in cases that we have not seen any visual examples of the novel categories.

We denote our knowledge graph by $G = (V, E)$, where $V$ and $E$ denote the nodes and the edges between nodes, respectively. Specifically, each node $v \in V$ denotes an object category, and each edge $e \in E$ denotes a relationship between a pair of object categories.

We use the Visual Genome (Krishna et al., 2017) dataset as a source to build the knowledge graph. Visual Genome consists of over 100K natural images. Each image is annotated with objects, attributes and the relationships between objects. Since there is no predefined object category list, the annotators are free to label any objects in the image, which results in very diverse object categories.

In our experiments, we build the knowledge graph by including all object categories that appear in the AI2-THOR environment. Each object category is represented as a node in the graph. We count the occurrence of object-to-object relationships in the Visual Genome dataset. Two nodes are connected with an edge only when the occurrence frequency of any relationship is more than three. Some examples of the mined relationships are shown in Figure 3.

### 4.2 INCORPORATING SEMANTIC KNOWLEDGE INTO ACTOR-CRITIC MODEL

The baseline policy model decides the action using the current state and target object features. However, we want the policy network to incorporate semantic knowledge of the world when planning the actions. How do we represent the semantic knowledge? More importantly, how do we extract semantic knowledge in the context of the current environment and state?

Our core idea is that the graph structure represents how the information propagates between different nodes. We initialize each node based on the current state (input scene image) and then perform information propagation to compute a semantic knowledge vector that is passed as another feature vector to the policy function. For information propagation, we use the recently proposed Graph Convolutional Network (GCN) (Kipf & Welling, 2017).

Figure 4: **Graph Convolutional Networks.** Each node denotes an object category and is initialized based on the the current state (image) and the word vector. We use three layers of GCN to perform information propagation. The first two layers output 1024-d latent features, and the last layer generates a single value for each node, which results in a $|V|$ dimensional semantic knowledge vector that is passed to the policy model.

### 4.2.1 GRAPH CONVOLUTIONAL NETWORK (GCN)

The GCNs are the extension of the Convolution Neural Networks to graph structures, where the goal is to learn a function representation for a given graph $G = (V, E)$. The input to each node $v$ is a feature vector $x_v$. We summarize the inputs of all nodes as a matrix $X = [x_1, \cdots, x_{|V|}] \in R^{|V| \times D}$, where $D$ denotes the dimension of the input feature. The graph structure is represented as a binary adjacency matrix $A$. We perform normalization on $A$ following (Kipf & Welling, 2017) and obtain $\widehat{A}$. The GCN outputs a node-level representation $Z = [z_1, \cdots, z_{|V|}] \in R^{|V| \times F}$. Let $f(\cdot)$ denote the ReLU activation function, we have

$$H^{(l+1)} = f(\widehat{A}H^{(l)}W^{(l)}) \tag{2}$$

with $H^{(0)} = X$ and $H^{(L)} = Z$, where $W^{(l)}$ is the parameter for the $l$-th layer and $L$ is the number of GCN layers.

### 4.2.2 GCN FOR NAVIGATION

In our visual semantic navigation task, the input of each node is designed as a joint representation of both the semantic cues (e.g., the word embedding) and the visual cues (e.g., the image classification score depending on the current state $s_t$). Specifically, the word embedding is generated by fastText (Joulin et al., 2016) and the classification score is generated by a ResNet-50 (He et al., 2016) pretrained on the 1000-class ImageNet dataset. Note that the classification score is obtained based on the frame of the current state and we have different word embeddings for different graph nodes. These two representations are first mapped to 512-d features by two different fully connected layers respectively. We then concatenate these two features and form a 1024-d joint representation for each graph node.

As illustrated in Figure 4, we use three layers of GCN, the first two layers output 1024 dimensional latent features, the last layer outputs a single value for each node which results in a $|V|$ dimensional feature vector. This feature vector is basically an encoding of semantic prior in the context of the current scene and environment.

Finally, we map this feature vector into the 512-d feature embedding and concatenate it with the features generated from the visual and semantic branches (1024-d embedding), which results in a 1536-d feature vector. As illustrated in Figure 2, the joint feature is further fed into the policy network for policy prediction.

## 5 EXPERIMENTS

In this section, we provide the results of navigation using GCNs. We evaluate our model in scenarios where the scenes are unseen and/or the target objects are novel to the agent. We also provide ablation results that show the knowledge graph is useful.

### 5.1 EVALUATION FRAMEWORK

We evaluate our method in the interactive environments of AI2-THOR (Kolve et al., 2017). AI2-THOR provides 120 scenes covering four different room categories: *kitchens*, *living rooms*, *bedrooms*, and *bathrooms*. Each room category consists of 30 rooms with diverse appearance and configurations. We randomly split the scenes into three splits for each room category, i.e., 20 training rooms, 5 validation rooms, and 5 testing rooms.

|  |  | Kitchen | Living room | Bedroom | Bathroom | Avg. |
|---|---|---|---|---|---|---|
| Seen scenes, | Random | 2.4 / 3.5 | 1.1 / 1.7 | 1.8 / 2.7 | 3.2 / 4.8 | 2.1 / 3.1 |
|  | A3C | 38.5 / 51.0 | 9.7 / 15.1 | 6.8 / 11.5 | 69.1 / 81.0 | 31.1 / 39.6 |
| Known objects | Ours | **58.6 / 72.7** | **12.4 / 18.6** | **41.6 / 52.4** | **71.3 / 83.0** | **46.0 / 56.7** |
| Seen scenes, | Random | 0.9 / 1.3 | 0.8 / 1.2 | 2.3 / 3.4 | 1.4 / 2.1 | 1.4 / 2.0 |
|  | A3C | 2.1 / 4.9 | 3.2 / 4.8 | 0.5 / 1.7 | 17.1 / 28.5 | 5.7 / 9.9 |
| **Novel** objects | Ours | **3.2 / 6.1** | **9.8 / 16.2** | **6.2 / 8.6** | **24.7 / 37.3** | **11.0 / 17.1** |
| **Unseen** scenes, | Random | 4.1 / 5.9 | 0.9 / 1.3 | 1.6 / 2.4 | 4.2 / 6.2 | 2.7 / 3.9 |
|  | A3C | 11.5 / 18.8 | 0.5 / 2.5 | 2.2 / 3.8 | 8.6 / 18.7 | 5.7 / 10.4 |
| Known objects | Ours | **12.7 / 20.5** | **1.0 / 4.0** | **4.5 / 11.0** | **8.7 / 21.1** | **6.7 / 13.4** |
| **Unseen** scenes, | Random | 2.0 / 2.8 | 0.6 / 1.0 | **2.0** / 2.8 | 2.7 / 3.9 | 1.8 / 2.6 |
|  | A3C | 2.2 / 7.5 | 2.5 / 4.4 | 1.3 / 4.4 | 3.4 / 9.3 | 2.4 / 5.9 |
| **Novel** objects | Ours | **3.3 / 12.7** | **2.8 / 5.3** | **2.0 / 6.3** | **4.1 / 12.2** | **3.1 / 8.5** |

Table 1: **Results using termination (stop) action.** SPL / Success rate (%) is shown. We compare against a random baseline and A3C (Mnih et al., 2016).

There are 87 object categories within AI2-THOR that are common among the scenes. However, some of the objects are not visible without interaction. For example, spoons were not visible since they always appeared in closed drawers during random initialization of the scenes so we did not use spoon among our categories. Therefore, we have $|V| = 53$ categories based on their visibility at random initialization of the scenes. To test the generalization ability of our method on novel objects, we split the 53 object categories into known and novel sets. Only the known set of object categories are used in training. The full split of object categories is shown in Appendix A. We only use navigation commands of AI2-THOR for our experiments. These actions include: *move forward*, *move back*, *rotate right*, *rotate left*, and *stop*.

We evaluate the models based on two metrics: *Success Rate* and the *Success weighted by Path Length (SPL)* metric recently proposed by Anderson et al. (2018a). *Success Rate* is defined as the ratio of the number of times the agent successfully navigates to the target and the total number of episodes. *SPL* is a better metric which is a function considering both *Success Rate* and the path length to reach the goal from the starting point. It is defined as $\frac{1}{N} \sum_{i=1}^{N} S_i \frac{L_i}{\max{(P_i, L_i)}}$, where $N$ is the number of episodes, $S_i$ is a binary indicator of success in episode $i$, $P_i$ represents path length and $L_i$ is the shortest path distance (provided by the environment for evaluation) in episode $i$.

## 5.2 RESULTS

We train each of the models three times with different random initializations. We show the training curves in Appendix B, where we plot the curves with error bands representing the standard deviation. The curves show that our proposed model converges in fewer training episodes compared to baseline and achieves better *Success Rate* as well as *SPL*, which shows the effectiveness of scene priors.

For evaluation, we run 250 episodes for each scene, where the initial location and orientation of the agent is randomized. The target object is randomly sampled for each episode. We select the models which perform best on the validation set for all methods and evaluate them on the test set.

We compare the performance of the following models: (1) **Random walk**, which is the simplest baseline for navigation. The agent randomly samples an action from the action space at each step. (2) **A3C (Mnih et al., 2016)**, which refers to the baseline model presented in Section 3.2. It is a state-of-the-art deep reinforcement learning model. (3) **Ours**, which is our proposed model. Each node of the first layer of GCNs is fed by a joint representation of the word embedding and the image classification scores extracted by ResNet-50, which depends on the current observed image.

We analyze the generalization ability of our method for unseen scenes and novel objects. Specifically, there are three experimental settings: 1) test on seen scenes with novel object categories as the navigation target; 2) test on unseen scenes with known object categories; and 3) test on unseen scenes with novel object categories. Table 1 shows the results for these different settings. In addition to the above settings, we also provide the results for seen scenes and known objects in the first row of the table. Note that most previous work (e.g., Zhu et al. (2017)) assume the environment notifies the agent when it reaches the target, and the agent does not have any idea if it has reached the target or not. In contrast, we consider the *stop* action and expect the agent to issue this action when it reaches the target. As mentioned in Section 3.2, this makes the learning challenging. In Table 2, we report the results for the simpler case where we remove the "stop" action from the list of actions.

|  |  | Kitchen | Living room | Bedroom | Bathroom | Avg. |
|---|---|---|---|---|---|---|
| Seen scenes, Known objects | Random | 17.9 / 33.1 | 12.1 / 30.5 | 16.8 / 51.2 | 24.5 / 34.6 | 17.8 / 37.3 |
| | A3C | 79.9 / 86.7 | 38.8 / 57.6 | 87.8 / 89.5 | **93.7 / 96.6** | 75.0 / 82.5 |
| | Ours | **83.5 / 88.2** | **46.4 /64.4** | **90.6 / 92.7** | 93.6 / 96.5 | **78.5 / 85.5** |
| Seen scenes, **Novel** objects | Random | 10.0 / 23.1 | 8.0 / 18.5 | 17.3 / 35.2 | 11.2 / 32.2 | 11.6 / 27.2 |
| | A3C | 20.2 / 38.8 | 24.2 / 46.5 | 23.5 / 35.8 | 50.2 / 74.6 | 29.5 / 48.9 |
| | Ours | **22.9 / 53.6** | **39.5 / 66.5** | **26.1 / 38.9** | **50.5 / 78.6** | **34.7 / 59.4** |
| **Unseen** scenes, Known objects | Random | 27.3 / 45.2 | 5.6 / 16.6 | 13.1 / 34.5 | 36.0 / 49.1 | 20.5 / 36.3 |
| | A3C | 39.5 / 56.2 | 12.0 / 31.8 | 22.5 / 49.2 | 47.4 / 60.2 | 30.3 / 49.3 |
| | Ours | **46.2 / 62.5** | **13.8 / 40.6** | **26.5 / 58.6** | **51.5 / 65.8** | **34.5 / 56.9** |
| **Unseen** scenes, **Novel** objects | Random | 21.3 / 44.3 | 3.3 / 22.9 | 25.8 / 47.8 | 25.5 / 48.9 | 19.0 / 41.0 |
| | A3C | 26.1 / 56.3 | 9.4 / 25.1 | 28.2 / 54.0 | 33.8 / 90.7 | 24.4 / 56.5 |
| | Ours | **38.5 / 62.5** | **13.7 / 40.3** | **30.1 / 63.1** | **39.2 / 93.6** | **30.4 / 64.9** |

Table 2: **Results without termination (stop) action.** SPL / Success rate (%) is shown. We compare against a random baseline and A3C. This scenario is simpler than the case shown in Table 1.

Our method that incorporates the knowledge graph outperforms the baselines in terms of both success rate and SPL. We observe a higher performance for the case that we do not use a *stop* action (Table 2), which is expected. The scenario in which both scenes and target objects are novel is quite challenging, and the performance degrades drastically for both A3C and our method. However, the performance is significantly better than random. The bathroom scenes are typically small so there is not much difference between the performance of our method and the baseline. Note that more than half of the object categories are not among ImageNet categories. Also, note that "Unseen scenes, Novel objects" is not necessarily the hardest case. For instance, in "Seen scenes, Novel objects", the appearance of the object and the mapping between the name and the object appearance are still unknown. We also observe overfitting to known scenes and objects (refer to "Seen scenes, Known objects"). So the results of different cases are not directly comparable, and it depends on the structure of the scenes and the configuration of objects. We show some qualitative examples in Appendix D, and the implementation details are provided in Appendix C.

**Generalization Across Scene Types.** We evaluate generalization across scene types as well. The idea is that we train the model on one scene type and evaluate it on a different scene type. The result is close to random in the scenario with the termination action. This is expected since there are very few common objects among different scene categories. The result for the simpler case of without the termination action is shown in Table 3.

|  |  | Test type | | | |
|---|---|---|---|---|---|
|  |  | Kitchen | Living room | Bedroom | Bathroom |
| Train type | Kitchen | 38.5 / 62.5 | 4.5 / 8.1 | 28.2 / 52.4 | 31.7 / 66.7 |
| | Living room | 22.6 / 52.1 | 13.7 / 40.3 | 27.0 / 48.0 | 26.9 / 60.1 |
| | Bedroom | 29.5 / 58.4 | 10.4 / 30.1 | 30.1 / 63.1 | 28.0 / 55.1 |
| | Bathroom | 35.4 / 71.9 | 5.9 / 17.9 | 24.1 / 35.8 | 39.2 / 93.6 |

Table 3: Results of generalization across scene types. SPL / Success rate (%) is shown.

**Ablations on Knowledge Graph.** We perform evaluations on how the performance is affected by changing the knowledge graph in our model. The experiment is performed with the *kitchen* scenes without the "stop" action. We first remove different fractions of object nodes or relations from the graph and re-train the models. As shown in Table 4, the SPL performance drops as more information is removed from the knowledge graph. We also train our model with a fully-connected graph which leads to the SPL of 32.5 and the model with a random graph leads to the SPL of $30.1 \pm 0.6$ (we repeated this experiment three times). The performance of these two cases is worse than the performance of the model with a proper knowledge graph (38.5).

| Drop % | 0% | 20% | 40% | 60% | 80% |
|---|---|---|---|---|---|
| Objects | 38.5 | 34.8 | 33.7 | 33.5 | 31.1 |
| Relations | 38.5 | 36.7 | 35.0 | 34.2 | 31.5 |

Table 4: Results of removing objects and relations in the knowledge graph.

We have tried using the edge types ("on", "next to", etc.), but the results is not better than the case that we ignore the edge types. That is probably due to the lack of training data for each type separately. We have also tried training only one model for all scene categories, but the performance is lower.

**Computation Cost.** It is worth mentioning that the GCN module in our model increases only $0.12$ GFLOPs computation compared to the baseline *A3C* ($\sim 4$ GFLOPs), which is marginal.

## 6 CONCLUSIONS

We propose an approach to integrate semantic and functional priors with a deep reinforcement learning model for the task of navigation. We use Graph Convolutional Networks to encode the prior knowledge and to update the knowledge according to the observations from the current scene. Our experiments show that prior knowledge improves generalization to unseen scenes and targets.

The current formulation of the problem does not include a long-term memory so in the future we plan to integrate memory to learn more complex exploration strategies. Incorporating higher-order relationships between objects and scenes is another future direction that we consider.

**Acknowledgements:** This research is partly sponsored by Google Focused Award and the ARO under Grant Number W911NF-18-1-0019. Abhinav was supported in part by Okawa Foundation. The views and conclusions contained in this document are those of the authors and should not be interpreted as representing the official policies, either expressed or implied, of the ARO or the U.S. Government. The U.S. Government is authorized to reproduce and distribute reprints for Government purposes notwithstanding any copyright notation herein.

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

## APPENDIX A    NAVIGATION TARGETS

In Table 5, we show the object categories that are used as our navigation targets. The split of train and test categories is provided as well.

| Room type | Train objects | Test objects |
|---|---|---|
| Kitchen | HousePlant, StoveKnob, Sink, TableTop, Potato, Bread, Tomato, Knife, Cabinet, Fridge, Container, ButterKnife, Lettuce, Pan, Bowl, CoffeeMachine, StoveBurner, Plate | Mug, Apple, Microwave, Toaster |
| Living room | Television, HousePlant, Chair, TableTop, Box, Cloth, Newspaper, KeyChain, WateringCan, PaintingHanger | Painting, Statue |
| Bedroom | Painting, HousePlant, CellPhone, LightSwitch, Candle, TableTop, Bed, Lamp, Statue, Book, CreditCard, KeyChain, Bowl, Pen, Box, Pencil, Blinds, Laptop, AlarmClock | Television, Mirror, Cabinet |
| Bathroom | SprayBottle, Painting, Candle, LightSwitch, Sink, Cabinet, TowelHolder, Watch, ToiletPaper, ShowerDoor, SoapBottle | SoapBar, Towel |

Table 5: Training and testing split of object categories for each scene type in the AI2-THOR.

## APPENDIX B    TRAINING CURVES

We show the training curves in Figure 5. We compare our method with the baseline A3C. All the models are trained 3 times with different initializations. We compute the model performance with *Success Rate* and *SPL* every 10 million iterations during training. We use the error band to represent the standard deviation. The curves show our model converges faster than the A3C baseline and obtain better performance in both metrics, which indicates the effectiveness of the scene priors.

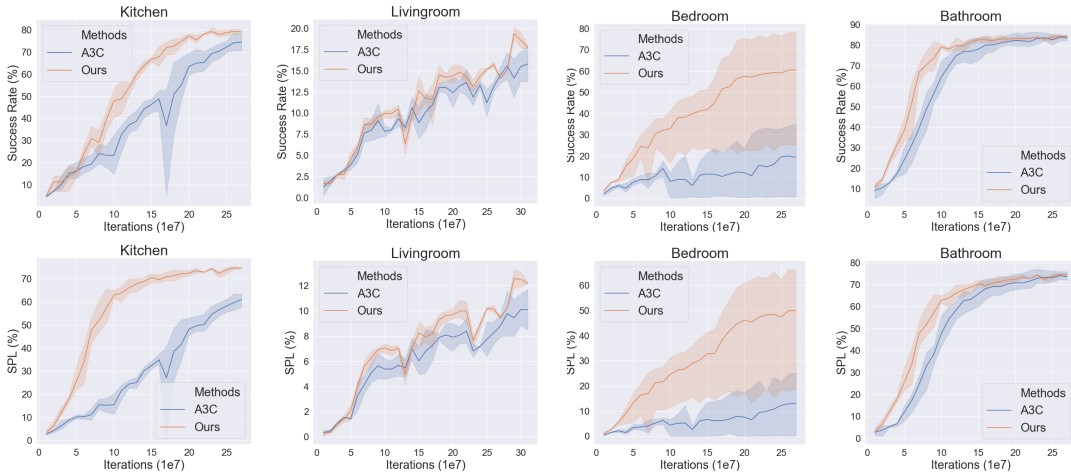

Figure 5: **Learning curves**. The top row shows *success rate* and the bottom row shows *SPL*.

## APPENDIX C    IMPLEMENTATION DETAILS

Our method is implemented in Tensorflow (Abadi et al., 2015) and the actor-critic policy network is trained with a single NVIDIA GeForce GTX Titan X GPU with 20 threads for 10 million frames for experiments without *stop* action, and for 25 million frames for experiments with *stop* action. The initial learning rate is set empirically as $7e - 4$, and is decreased linearly as the training progresses. The network parameters are optimized by the RMSProp optimizer (Tieleman & Hinton). The maximum number of steps is set to 100 for *kitchen*, *bedroom* and *bathroom*, and to 200 for *living room* due to the larger exploration space. Since there is almost no overlap between object categories within different room types, we train separate models for each room type.

## APPENDIX D    QUALITATIVE RESULTS

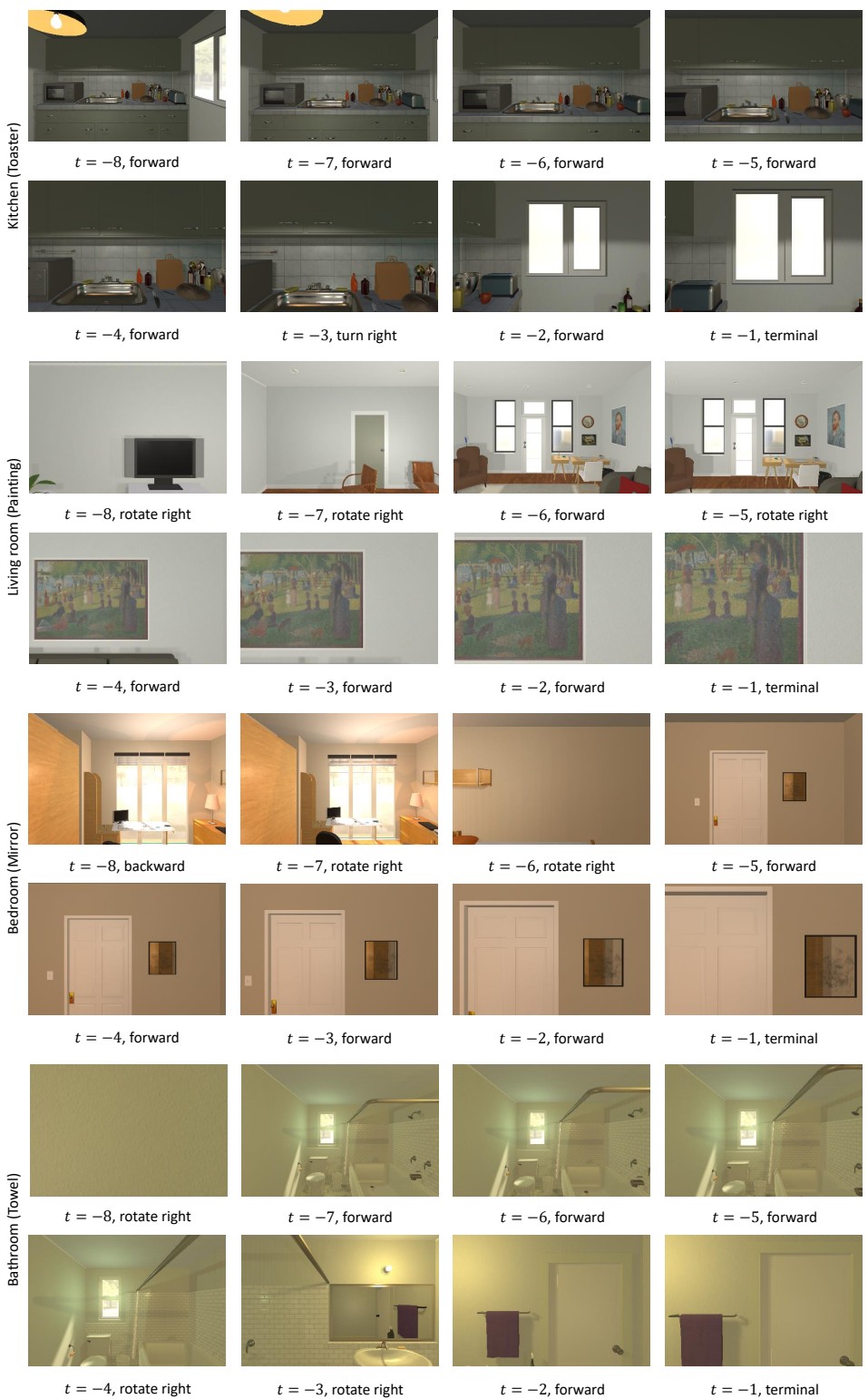

Figure 6: **Qualitative results.** Examples of last eight frames and the corresponding actions $a_t$ predicted from our model on unseen scenes with novel target objects.

