# OpenReview forum: "Visual Semantic Navigation using Scene Priors"
_ICLR.cc/2019/Conference_

### Official Review · AnonReviewer1 · 2018-10-19
**Good paper, some additional experiments would make it stronger**

**Rating:** 7
**Confidence:** 4

**Review:**

This work proposes to use semantic knowledge about the relationships and functionality of different objects, to help in navigation tasks, in both familiar and unfamiliar situations. The paper is very well written and it is clear what the authors did. The approach seems sound, and while it combines two existing approaches (actor-critic reinforcement learning for navigation, and belief propagation using graph convolution networks) is sufficiently novel to be of interest to at least some members of the community. The experimental evaluation is good, and the proposed method outperforms Mnih 2016 by a significant margin, especially in the more interesting settings. A good ablation study is provided.

My main concern is that there seems to be a larger pool of work in semantic navigation than what the evaluation includes. Anderson 2018, Zhu 2017 and Gupta 2017 seem relevant. While none of these use knowledge graphs, some of these show they outperform Mnih 2016 so would be stronger baselines.

I am also curious whether the proposed work generalizes across scene type categories (e.g. if it learns on kitchens but it tested on living rooms). This would be an experiment in the spirit of unknown object/scene but even more challenging.

---

> ### Author Response · Authors · 2018-11-19
> **Response to Reviewer 1**
>
> Thank you for the insightful feedback. Please find the answers to the questions and comments below.
>
> - Comparison with Anderson 2018, Zhu 2017 and Gupta 2017
>
> Answer: Zhu et al., 2017 use the picture of the target as input, while we consider scenarios with unseen objects. Also, they train and test in the same scenes, and we use unseen scenes and targets for evaluation. Gupta et al.,  2017 train their model using imitation learning (DAGGER), which means they have the optimal action. We use only the scalar reward for completing the task. Regarding Anderson et al., 2018, the agent receives natural language instructions, while we use a single word specifying the target category. Due to these discrepancies, we cannot make an apples to apples comparison.
>
> - I am also curious whether the proposed work generalizes across scene type categories.
>
> Answer: Thanks for suggesting this experiment. We have added a new paragraph in the result section to describe this experiment. We also added a new table to the revised version (Table 3), where we show the results for training on one category and testing on another category.

---

> > ### Comment · AnonReviewer1 · 2018-12-03
> > **thanks**
> >
> > Thanks for the clarification regarding potential baselines, and for the additional results. It looks like in most cases the method's performance degrades gracefully as you test on different scene categories than what was used for training. I guess I was really wondering how the method would fare on this task *compared to baselines* but I didn't explicitly state this request so that's ok. I retain my positive rating.

---

### Official Review · AnonReviewer3 · 2018-10-28
**fine paper; some questions**

**Rating:** 7
**Confidence:** 3

**Review:**

This paper explores the use of semantic priors for semantic navigation. The semantic priors are derived from language datasets (in the form of word embeddings, which assign similar feature vectors to related words) and from visual datasets (Visual Genome, which represents relationships between objects that co-occur in scenes).

The general ideas are reasonable. The experimental protocol is sound and uses recent best practices. The results are fine.

I'm a bit puzzled by the way the GCN is used. Figure 2 implies that the GCN doesn't actually use information from the current image. I.e., the GCN input doesn't change as the agent navigates the scene. (In Figure 2, the GCN path appears similar to the Word Embedding path. The Word Embedding path doesn't update when the agent moves, so the reader can infer that the GCN path doesn't update either.) But then I don't quite understand how the GCN incorporates information from the current scene.

Figure 4 implies that the GCN is re-evaluated when the agent moves and the input image changes. But how is information from the image fed into the GCN? The text implies that an ImageNet classification model is run on the image. But why image classification and not object detection? It seems that what one would really want is to understand what objects are in the scene. And how is output of the image classification network supplied to the GCN? Is the target object type used as well? Overall it's not clear to me exactly how information from the current image is supplied to the GCN, why this mechanism is right, and what the GCN is expected to do. (I do understand how GCNs work, just not how exactly they are used here and why this precise usage is right for this application.) I hope the authors can clarify in the response.

---

> ### Author Response · Authors · 2018-11-19
> **Response to Reviewer 3**
>
> We appreciate the insightful feedback. We have addressed the questions and comments below.
>
> - Figure 2 implies that the GCN doesn't actually use information from the current image.
>
> Answer: We have modified Figure 2 in the revision for clarification. The GCN actually takes the information from the current image by computing the 1000-class ImageNet classification score on it. This 1000-d score is forwarded to an FC layer which outputs a 512-d image embedding. For each node, this 512-d image embedding is concatenated with a word embedding (512-d), which creates a 1024-d feature for input. Note that the word embeddings are different corresponding to the semantic class for each node.
>
> - Why image classification and not object detection?
>
> Answer: We agree that a detector might be better. That said, we did try to apply Faster R-CNN trained with COCO. However, the Faster RCNN detector generated a lot of false detections and localizations. The benefit of using ImageNet classifier is that it gives relatively more robust estimation as it does not need to handle localization. Moreover, prediction of more diverse classes allows more complex relationship reasoning.
>
>
> - What the GCN is expected to do?
>
> Answer: The input for each node in the GCN changes after every action based on the new observed image. The input for each node is the joint embedding of the current observation (image embedding) and the semantic class of the node (word embedding). By propagating the information through the edges of the knowledge graph, the information for each node is updated by its related nodes. Intuitively, in this way, the information of the existence of a “coffee machine” can be propagated to highlight the potential existence of a “mug”. We concatenate the response of all nodes to form a feature for policy estimation. This helps us generalize to navigation to “mug”, although we do not optimize directly for it during training.

---

> ### Comment · AnonReviewer3 · 2018-12-09
> **vote to accept**
>
> I upgraded my score from 6 to 7.
>
> The authors have responded satisfactorily to my questions. I still find the method a bit convoluted and do not think that it will stand the test of time. However, the paper is competently done and is a fine addition to the literature. I support acceptance.

---

### Official Review · AnonReviewer2 · 2018-10-29
**Visual Semantic Navigation using Scene Priors**

**Rating:** 7
**Confidence:** 1

**Review:**

This paper tackles the problem of navigating scenes to find objects which are potentially not included in the training phase. To find an unseen object from a scene, the proposed model incorporates an external knowledge graph as an augmented input of the actor-critic model. To construct a knowledge graph, entities in a scene are identified by ResNet and then the link structure between entities are extracted from VIsual Genome dataset. Through the ablation study, it is shown that using the knowledge graph helps to track and identify unseen objects during training.

- The original knowledge graph (KG) has relation labels (such as next to, on in figure 3) between different objects, however, GCN does not take into account the relations between objects. Only co-occurrence patterns will be encoded into the KG constructed from an image. There are more complex graph convolutional models modelling relations between nodes such as [1]. Have you considered adding explicit relations between entities? will it increase the navigation performance? if not why?
- It is unclear how many objects are used to construct a KG from an image. For example, are top-k objects identified by ResNet used to construct a KG?
- Description of the reward is a bit unclear as well, especially when the model is trained without stop action. From the text, the agents receive a positive reward when it is close to the target (within a certain number of steps). Does this mean that the agent gets a positive reward on every step near the target while it's not in the final state?
- This might be a trivial question, but I couldn't find it from the text. Can you find all object from AI2-THOR in the categories of ImageNet and of Visual Genome? is there any information loss while constructing a KG from the classification result? What is the average number of nodes of a KG? and is there any correlation between the size of KG and the result?
- Why are the performances of the models is unstable with Bedroom dataset (in terms of variance)?
- The input feature of GCN is a combination of word feature and image feature. It is clear that there is a corresponding word embedding for each of the identified objects, but it is unclear what is the corresponding image feature. If two objects are identified in the same frame, do input features of these two objects share the same image features from Resnet?

[1] Schlichtkrull, Michael, et al. "Modeling relational data with graph convolutional networks." European Semantic Web Conference. Springer, Cham, 2018.

---

> ### Author Response · Authors · 2018-11-19
> **Response to Reviewer 2**
>
> Thank you for the valuable comments and clarifying questions. Please find the responses to your questions and comments below.
>
> - Have you considered adding explicit relations between entities? will it increase the navigation performance? if not why?
>
> Answer: Yes, we used the explicit relations but it did not improve the results (we briefly mention that towards the end of Section 5.2). That is probably due to overfitting since we have few examples for each type of relation.
>
> - It is unclear how many objects are used to construct a KG from an image. For example, are top-k objects identified by ResNet used to construct a KG?
>
> Answer: The number of nodes is fixed and it is not image dependent. We are considering 53 objects of THOR so our graph has 53 nodes (Section 5.1).
>
> - The agents receive a positive reward when it is close to the target (within a certain number of steps). Does this mean that the agent gets a positive reward on every step near the target while it's not in the final state?
>
> Answer: In the scenario that we use the termination action, the agent should say “stop” when it observes the target object to get the reward. Otherwise, it will not receive the reward. In the scenario that we do not have the termination action, the agent might receive the positive reward at multiple points since we reward the agent if the target is within the cone of visibility and within 1 meter from the agent. Once it receives the reward the episode is finished.
>
> - Can you find all object from AI2-THOR in the categories of ImageNet and of Visual Genome? is there any information loss while constructing a KG from the classification result?
>
> Answer: About half of the object categories are not in ImageNet. However, all of them appear in Visual Genome.
>
> - What is the average number of nodes of a KG? and is there any correlation between the size of KG and the result?
>
> Answer: We use 53 nodes. In Table 3 of the original submission (Table 4 of the revised version), we show how the performance degrades as we remove nodes and relations from the graph.
>
> - Why are the performances of the models is unstable with Bedroom dataset (in terms of variance)?
>
> Answer: One run got stuck in a bad local minima and that caused a large variance. We have multiple runs with different random initialization.
>
> - It is unclear what is the corresponding image feature. If two objects are identified in the same frame, do input features of these two objects share the same image features from Resnet?
>
> Answer: Yes, that is right. We use image-level features (as opposed to object-level features). Some of our object categories are novel and unseen so we cannot train supervised detectors for them.

---

### Public Comment · ~Guyue_Hu2 · 2018-10-24
**What is the training pipeline?**

Hi,
I am not very clear about the training pipeline, In Fig.2, does all the embedding sub-networks(the Visual network, the Semantic network, and the Graph network) are simultaneously end-to-end optimized with Actor-critic model ? Or the  embedding sub-networks are frozen when training the policy network (Actor-cirtic model). Can you do me a help? Thank you!

---

> ### Author Response · Authors · 2018-10-25
> **training pipeline**
>
> The graph network is optimized end-to-end along with the actor-critic model. There is no backpropagation to the visual network and semantic network. The fully connected layers after them are trained along with the actor-critic model though.

---

### Meta-Review · Area_Chair1 · 2018-12-17

**Recommendation:** Accept (Poster)
**Confidence:** 4

**Metareview:**

The authors propose an approach for visual navigation that leverages a semantic knowledge graph to ground and inform the policy of an RL agent. The agent uses a graphnet to learn relationships and support the navigation. The empirical protocol is sound and uses best practices, and the authors have added additional experiments during the revision period, in response to the reviewers' requests. However, there were some significant problems with the submission - there were no comparisons to other semantic navigation methods, the approach is somewhat convoluted and will not survive the test of time, and the authors did not conclusively show the value of their approach. The reviewers uniformly support the publication of this paper, but with a low confidence.